# $A^2$-Nets: Double Attention Networks

**Yunpeng Chen**[*]
National University of Singapore
chenyunpeng@u.nus.edu

**Yannis Kalantidis**
Facebook Research
yannisk@fb.com

**Jianshu Li**
National University of Singapore
jianshu@u.nus.edu

**Shuicheng Yan**
Qihoo 360 AI Institute
National University of Singapore
eleyans@nus.edu.sg

**Jiashi Feng**
National University of Singapore
elefjia@nus.edu.sg

## Abstract

Learning to capture long-range relations is fundamental to image/video recognition. Existing CNN models generally rely on increasing depth to model such relations which is highly inefficient. In this work, we propose the "double attention block", a novel component that aggregates and propagates informative global features from the entire spatio-temporal space of input images/videos, enabling subsequent convolution layers to access features from the entire space efficiently. The component is designed with a double attention mechanism in two steps, where the first step gathers features from the entire space into a compact set through second-order attention pooling and the second step adaptively selects and distributes features to each location via another attention. The proposed double attention block is easy to adopt and can be plugged into existing deep neural networks conveniently. We conduct extensive ablation studies and experiments on both image and video recognition tasks for evaluating its performance. On the image recognition task, a ResNet-50 equipped with our double attention blocks outperforms a much larger ResNet-152 architecture on ImageNet-1k dataset with over $40\%$ less the number of parameters and less FLOPs. On the action recognition task, our proposed model achieves the state-of-the-art results on the Kinetics and UCF-101 datasets with significantly higher efficiency than recent works.

## 1 Introduction

Deep Convolutional Neural Networks (CNNs) have been successfully applied in image and video understanding during the past few years. Many new network topologies have been developed to alleviate optimization difficulties [9, 10] and increase the learning capacities [26, 5], which benefit recognition performance for both images [8, 2] and videos [23] significantly.

However, CNNs are inherently limited by their convolution operators which are dedicated to capturing local features and relations, *e.g.* from a $7 \times 7$ region, and are inefficient in modeling long-range interdependencies. Though stacking multiple convolution operators can enlarge the receptive field, it also comes with a number of unfavorable issues in practice. First, stacking multiple operators makes the model unnecessarily deep and large, resulting in higher computation and memory cost as well as increased over-fitting risks. Second, features far away from a specific location have to pass through a stack of layers before affecting the location for both forward propagation and backward propagation, increasing the optimization difficulties during the training. Third, the features visible to a distant location are actually "delayed" ones from several layers behind, causing inefficient reasoning. Though

---

[*]Part of the work is done during internship at Facebook Research.

some recent works [11, 25] can partially alleviate the above issues, they are either non-flexible [11] or computationally expensive [25].

In this work, we aim to overcome these limitations by introducing a new network component that enables a convolution layer to sense the entire spatio-temporal space[2] from its adjacent layer immediately. The core idea is to first *gather* key features from the entire space into a compact set and then *distribute* them to each location adaptively, so that the subsequent convolution layers can sense features from the entire space even without a large receptive filed. We develop a generic function for such purpose and implement it with an efficient *double attention* mechanism. The first second-order attention pooling operation selectively gathers key features from the entire space, while the second adopts another attention mechanism to adaptively distribute a subset of key features that are helpful to complement each spatio-temporal location for high-level tasks. We denote our proposed double-attention block as $A^2$-block and its resultant network as $A^2$-Net.

The double-attention block is related to a number of recent works, including the Squeeze-and-Excitation Networks [11], covariance pooling [14], the Non-local Neural Networks [25] and the Transformer architecture of [24]. However, compared with these existing works, it enjoys several unique advantages: Its first attention operation implicitly computes second-order statistics of pooled features and can capture complex appearance and motion correlations that cannot be captured by the global average pooling used in SENet [11]. Its second attention operation *adaptively* allocates features from a compact bag, which is more efficient than exhaustively correlating the features from all the locations with every specific location as in [25, 24]. Extensive experiments on image and video recognition tasks clearly validate the above advantages of our proposed method.

We summarize our contributions as follows:

- We propose a generic formulation for capturing long-range feature interdependencies via universal gathering and distribution functions.

- We propose the double attention block for gathering and distributing long-range features, an efficient architecture that captures second-order feature statistics and makes adaptive feature assignment. The block can model long-range interdependencies with a low computational and memory footprint and at the same time boost image/video recognition performance significantly.

- We investigate the effect of our proposed $A^2$-Net with extensive ablation studies and prove its superior performance through comparison with the state-of-the-arts on a number of public benchmarks for both image recognition and video action recognition tasks, including ImageNet-1k, Kinetics and UCF-101.

The rest of the paper is organized as follows. We first motivate and present our approach in Section 2, where we also discuss the relation of our approach to recent works. We then evaluate and report results in Section 3 and conclude the paper with Section 4.

## 2 Method

Convolutional operators are designed to focus on local neighborhoods and therefore fail to "sense" the entire spatial and/or temporal space, *e.g.* the entire input frame or one location across multiple frames. A CNN model thus usually employs multiple convolution layers (or recurrent units [6, 17]) in order to capture global aspects of the input. Meanwhile, self-attentive and correlation operators like second-order pooling have been recently shown to work well in a wide range of tasks [24, 14, 15]. In this section we present a component capable of gathering and distributing global features to each spatial-temporal location of the input, helping subsequent convolution layers sense the entire space immediately and capture complex relations. We first formally describe this desired component by providing a generic formulation and then introduce our double attention block, a highly efficient instantiation of such a component. We finally discuss the relation of our approach to other recent related approaches.

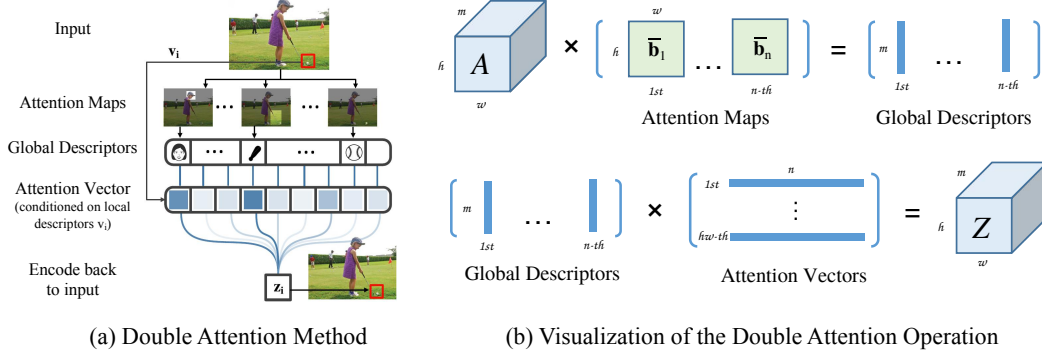

(a) Double Attention Method      (b) Visualization of the Double Attention Operation

Figure 1: Illustration of the double-attention mechanism. (a) An example on a single frame input for explaining the idea of our double attention method, where the set of global featues is computed only once and then shared by all locations. Meanwhile, each location $i$ will generate its own attention vector based on the need of its local feature $\mathbf{v}_i$ to select a desired subset of global features that is helpful to complement current location and form the feature $\mathbf{z}_i$. (b) The double attention operation on a three dimensional input array $A$. The first attention step is shown on the top and produces a set of global features. At location $i$, the second attention step generates the new local feature $\mathbf{z}_i$, as shown at the bottom.

Let $X \in \mathbb{R}^{c \times d \times h \times w}$ denote the input tensor for a spatio-temporal (3D) convolutional layer, where $c$ denotes the number of channels, $d$ denotes the temporal dimension[3] and $h$, $w$ are the spatial dimensions of the input frames. For every spatio-temporal input location $i = 1, \ldots, dhw$ with local feature $\mathbf{v}_i$, let us define

$$\mathbf{z}_i = \mathbf{F}_{\text{distr}}\left(\mathbf{G}_{\text{gather}}(X), \mathbf{v}_i\right), \tag{1}$$

to be the output of an operator that first *gathers* features in the entire space and then *distributes* them back to each input location $i$, taking into account the local feature $\mathbf{v}_i$ of that location. Specifically, $\mathbf{G}_{\text{gather}}$ adaptively aggregates features from the entire input space, and $\mathbf{F}_{\text{distr}}$ distributes the gathered information to each location $i$, conditioned on the local feature vector $\mathbf{v}_i$.

The idea of gathering and distributing information is motivated by the squeeze-and-excitation network (SENet) [11]. Eqn. (1), however, presents it in a more general form that leads to some interesting insights and optimizations. In [11], global average pooling is used in the gathering process, while the resulted single global feature is distributed to all locations, ignoring different needs across locations. Seeing these shortcomings, we introduce this genetic formulation and propose the *Double Attention block*, where global information is first gathered by second-order attention pooling (instead of first-order average pooling), and the gathered global features are adaptively distributed conditioned on the need of current local feature $\mathbf{v}_i$, by a second attention mechanism. In this way, more complex global relations can be captured by a compact set of features and each location can receive its customized global information that is complementary to the exiting local features, facilitating learning more complex relations. The proposed component is illustrated in Figure 1 (a). At below, we first describe its architecture in details and then discuss some instantiations and its connections to other recent related approaches.

## 2.1 The First Attention Step: Feature Gathering

A recent work [15] used bilinear pooling to capture second-order statistics of features and generate global representations. Compared with the conventional average and max pooling which only compute first-order statistics, bilinear pooling can capture and preserve complex relations better. Concretely, bilinear pooling gives a sum pooling of second-order features from the *outer product* of all the feature vector pairs $(\mathbf{a}_i, \mathbf{b}_i)$ within two input feature maps $A$ and $B$:

$$\mathbf{G}_{\text{bilinear}}(A, B) = AB^\top = \sum_{\forall i} \mathbf{a}_i \mathbf{b}_i^\top, \tag{2}$$

where $A = [\mathbf{a}_1, \cdots, \mathbf{a}_{dhw}] \in \mathbb{R}^{m \times dhw}$ and $B = [\mathbf{b}_1, \cdots, \mathbf{b}_{dhw}] \in \mathbb{R}^{n \times dhw}$. In CNNs, $A$ and $B$ can be the feature maps from the same layer, *i.e.* $A = B$, or from two different layers, *i.e.* $A = \phi(X; W_\phi)$ and $B = \theta(X; W_\theta)$, with parameters $W_\phi$ and $W_\theta$.

By introducing the output variable $G = [\mathbf{g}_1, \cdots, \mathbf{g}_n] \in \mathbb{R}^{m \times n}$ of the bilinear pooling and rewriting the second feature $B$ as $B = [\bar{\mathbf{b}}_1; \cdots; \bar{\mathbf{b}}_n]$ where each $\bar{\mathbf{b}}_i$ is a $dhw$-dimensional row vector, we can reformulate Eqn. (2) as

$$\mathbf{g}_i = A\bar{\mathbf{b}}_i^\top = \sum_{\forall j} \bar{\mathbf{b}}_{ij}\mathbf{a}_j. \tag{3}$$

Eqn. (3) gives a new perspective on the bilinear pooling result: instead of just computing second-order statistics, the output of bilinear pooling $G$ is actually a bag of visual primitives, where each primitive $\mathbf{g}_i$ is calculated by gathering local features weighted by $\bar{\mathbf{b}}_i$. This inspires us to develop a new attention-based feature gathering operation. We further apply a $\mathrm{softmax}$ onto $B$ to ensure $\sum_j \bar{\mathbf{b}}_{ij} = 1$, *i.e.* a valid attention weighting vector, which gives following *second-order attention pooling* process:

$$\mathbf{g}_i = A \, \mathrm{softmax}(\bar{\mathbf{b}}_i)^\top. \tag{4}$$

The first row in Figure 1 (b) shows the second-order attention pooling that corresponds to Eqn. (4), where both $A$ and $B$ are outputs of two different convolution layers transforming the input $X$. In implementation, we let $A = \phi(X; W_\phi)$ and $B = \mathrm{softmax}(\theta(X; W_\theta))$. The second-order attention pooling offers an effective way to gather key features: it captures the global features, *e.g.* texture and lighting, when $\bar{\mathbf{b}}_i$ is densely attended on all locations; and it captures the existence of specific semantic, *e.g.* an object and parts, when $\bar{\mathbf{b}}_i$ is sparsely attended on a specific region. We note that similar understandings were presented in [7], in which they proposed a rank-1 approximation of a bilinear pooling operation associated with a fully connected classifier. However, in our work, we propose to apply attention pooling to gather visual primitives at different locations into a bag of global descriptors using $\mathrm{softmax}$ attention map and do not apply any low-rank constraint.

## 2.2 The Second Attention Step: Feature Distribution

The next step after gathering features from the entire space is to distribute them to each location of the input, such that the subsequent convolution layer can sense the global information even with a small convolutional kernel.

Instead of distributing the same summarized global features to all locations like SENet [11], we propose to get more flexibility by distributing an adaptive bag of visual primitives based on the need of feature $\mathbf{v}_i$ at each location. In this way, each location can select features that are complementary to the current feature which can make the training easier and help capture more complex relations. This is achieved by selecting a subset of feature vectors from $\mathbf{G}_{\mathrm{gather}}(X)$ with soft attention:

$$\mathbf{z}_i = \sum_{\forall j} \mathbf{v}_{ij}\mathbf{g}_j = \mathbf{G}_{\mathrm{gather}}(X)\mathbf{v}_i, \ \ \text{where} \ \sum_{\forall j} \mathbf{v}_{ij} = 1. \tag{5}$$

Eqn. (5) formulates the proposed soft attention for feature selection. In our implementation, we apply the $\mathrm{softmax}$ function to normalize $\mathbf{v}_i$ into the one with unit sum, which is found to give better convergence. The second row in Figure 1 (b) shows the above feature selection step. Similar to the way we generate the attention map, the set of attention weight vectors is also generated by a convolution layer follow by a $\mathrm{softmax}$ normalizer, *i.e.* $\mathbf{V} = \mathrm{softmax}(\rho(X; W_\rho))$ where $W_\rho$ contains parameters for this layer.

## 2.3 The Double Attention Block

We combine the above two attention steps to form our proposed *double-attention block*, with its computation graph in deep neural networks is given in Figure 2. To formulate the double attention operation, we substitute Eqn. (4) and Eqn. (5) into Eqn. (1) and obtain

$$\begin{aligned} Z &= \mathbf{F}_{\mathrm{distr}}\left(\mathbf{G}_{\mathrm{gather}}(X), V\right) \\ &= \mathbf{G}_{\mathrm{gather}}(X)\mathrm{softmax}\left(\rho(X; W_\rho)\right) \\ &= \left[\phi(X; W_\phi)\mathrm{softmax}\left(\theta(X; W_\theta)\right)^\top\right]\mathrm{softmax}\left(\rho(X; W_\rho)\right). \end{aligned} \tag{6}$$

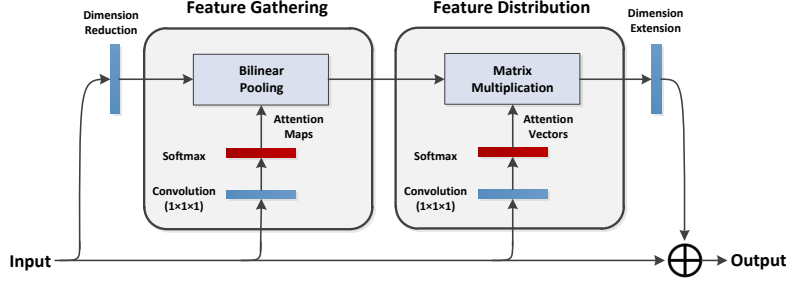

Figure 2: The computational graph of the proposed double attention block. All convolution kernel size is $1 \times 1 \times 1$. We insert this double attention block to existing convolutional neural network, *e.g.* residual networks [9], to form the $A^2$-Net.

Figure 1 (b) shows the combined double attention operation and Figure 2 shows the corresponding computational graph, where the feature arrays $A$, $B$ and $V$ are generated by three different convolution layers operating on the input feature array $X$ followed by softmax normalization if necessary. The output result $Z$ is given by conducting two matrix multiplications with necessary reshape and transpose operations. Here, an additional convolution layer is added at the end to expand the number of channels for the output $Z$, such that it can be encoded back to the input $X$ via element-wise addition. During the training process, gradient of the loss function can be easily computed using auto-gradient [3, 18] with the chain rule.

There are two different ways to implement the computational graph of Eqn. (6). One is to use the left association as given in Eqn. (6) with computation graph is shown in Figure 2. The other is to conduct the right association, as formulate below:

$$Z = \phi(X; W_\phi) \left[ \text{softmax} \left( \theta(X; W_\theta) \right)^\top \text{softmax} \left( \rho(X; W_\rho) \right) \right]. \qquad (7)$$

We note these two different associations are mathematically equivalent and thus will produce the same output. However, they have different computational cost and memory consumption. The computational complexity of the second matrix multiplication in "left association" in Eqn. (6) is $\mathcal{O}(mndhw)$, while "right association" in Eqn. (7) has complexity of $\mathcal{O}(m(dhw)^2)$. As for the memory cost[4], storing the output of the results of the first matrix multiplication costs $mn/2^{18}$MB and $(dhw)^2/2^{18}$MB for the left and right associations respectively. In practice, an input data array $X$ with 32 $28 \times 28$ frames and 512 channel size can easily cost more than 2GB memory when adopting the right association, much more expensive than 1MB cost of the left association. In this case, left association is also more computationally efficient than the right one. Therefore, for common cases where $(dhw)^2 > nm$, we suggest implementation in Eqn. (6) with left association.

## 2.4 Discussion

It is interesting to observe that the implementation in Eqn. (7) with right association can be further explained by the recent NL-Net [25], where the first multiplication captures pair-wise relations between local features and gives an output relation matrix in $\mathbb{R}^{dhw \times dhw}$. The resulted relation matrix is then applied to linearly combine the transformed features $\phi(X)$ into the output feature $Z$. The difference is apparent in the design of the pair-wise relation function, where we propose a new relation function, *i.e.* $\text{softmax} \left( \theta(X) \right)^\top \text{softmax} \left( \rho(X) \right)$ rather than using the Embedded Gaussian formulation [24] to capture the pair-wise relations. Meanwhile, as discussed above, any such a method practically suffers from high computational and memory costs, and relies on the some subsampling tricks to reduce the cost which may potentially hurts the accuracy. Since NL-Net is the current state-of-the-art for video recognition tasks and also closely related, we directly compare and extensively discuss performance between the two in the Experiments section. The results clearly show that our proposed method not only outperforms NL-Net, but does so with higher efficiency and accuracy. As the Embedded Gaussian NL-Net formulation that we compare in the experiments is mathematically equivalent to the self-attention formulation of [24], conclusions/comparisons to NL-Net extend to the transformer networks as well.

Table 1: Three backbone Residual Networks for the video tasks. The input size for ResNet-26 and ResNet-29 are $16\times112\times112$, while the input size for ResNet-50 is $8\times224\times224$. We follow [25] and set $k = [3,1,3],[3,1,3,1,3,1],[1,3,1]$ for ResNet-50 in last three stages and decrease the temporal size to reduce computational cost.

| stage | ResNet-26 | ResNet-29 | output | ResNet-50 | output |
|---|---|---|---|---|---|
| conv1 | $3\times5\times5$, 16, stride (1,2,2) | $3\times5\times5$, 16, stride (1,2,2) | $16\times56\times56$ | $3\times5\times5$, 32, stride (1,2,2) <br> max pooling, stride (1,2,2) | $8\times56\times56$ |
| conv2 | $\begin{bmatrix} 1\times1\times1,\ 32 \\ 3\times3\times3,\ 32 \\ 1\times1\times1,\ 128 \end{bmatrix}\times 2$ | $\begin{bmatrix} 1\times1\times1,\ 32 \\ 3\times3\times3,\ 32 \\ 1\times1\times1,\ 128 \end{bmatrix}\times 2$ | $8\times56\times56$ | $\begin{bmatrix} 1\times1\times1,\ \underline{64} \\ 3\times3\times3,\ \underline{64} \\ 1\times1\times1,\ \underline{256} \end{bmatrix}\times \underline{3}$ | $8\times56\times56$ |
| conv3 | $\begin{bmatrix} 1\times1\times1,\ 64 \\ 3\times3\times3,\ 64 \\ 1\times1\times1,\ 256 \end{bmatrix}\times 2$ | $\begin{bmatrix} 1\times1\times1,\ 64 \\ 3\times3\times3,\ 64 \\ 1\times1\times1,\ 256 \end{bmatrix}\times 2$ | $8\times28\times28$ | $\begin{bmatrix} 1\times1\times1,\ \underline{128} \\ k\times3\times3,\ \underline{128} \\ 1\times1\times1,\ \underline{512} \end{bmatrix}\times \underline{4}$ | $4\times28\times28$ |
| conv4 | $\begin{bmatrix} 1\times1\times1,\ 128 \\ 3\times3\times3,\ 128 \\ 1\times1\times1,\ 512 \end{bmatrix}\times 2$ | $\begin{bmatrix} 1\times1\times1,\ 128 \\ 3\times3\times3,\ 128 \\ 1\times1\times1,\ 512 \end{bmatrix}\times \underline{3}$ | $8\times14\times14$ | $\begin{bmatrix} 1\times1\times1,\ \underline{256} \\ k\times3\times3,\ \underline{256} \\ 1\times1\times1,\ \underline{1024} \end{bmatrix}\times \underline{6}$ | $4\times14\times14$ |
| conv5 | $\begin{bmatrix} 1\times1\times1,\ 256 \\ 3\times3\times3,\ 256 \\ 1\times1\times1,\ 1024 \end{bmatrix}\times 2$ | $\begin{bmatrix} 1\times1\times1,\ 256 \\ 3\times3\times3,\ 256 \\ 1\times1\times1,\ 1024 \end{bmatrix}\times 2$ | $8\times7\times7$ | $\begin{bmatrix} 1\times1\times1,\ \underline{512} \\ k\times3\times3,\ \underline{512} \\ 1\times1\times1,\ \underline{2048} \end{bmatrix}\times \underline{3}$ | $4\times7\times7$ |
| | global average pool, fc, softmax | global average pool, fc, softmax | $1\times1\times1$ | global average pool, fc, softmax | $1\times1\times1$ |
| (#Params, FLOPs) | (7.0 M, 8.3 G) | (7.6 M, 9.2 G) | | (33.4 M, 31.3 G) | |

# 3   Experiments

In this section, we first conduct extensive ablation studies to evaluate the proposed $A^2$-Nets on the Kinetics [12] video recognition dataset and compare it with the state-of-the-art NL-Net [25]. Then we conduct more experiments using deeper and wider neural networks on both image recognition and video recognition tasks and compare it with state-of-the-art methods.

## 3.1   Implementation Details

**Backbone CNN**   We use the residual network [10] as our backbone CNN for all experiments. Table 1 shows architecture details of the backbone CNNs for video recognition tasks, where we use ResNet-26 for all ablation studies and ResNet-29 as one of the baseline methods. The computational cost is measured by FLOPs, *i.e.* floating-point multiplication-adds, and the model complexity is measured by #Params, *i.e.* total number of trained parameters. The ResNet-50 is almost $2\times$ deeper and wider than the ResNet-26 and thus only used for last several experiments when comparing with the state-of-the-art methods. For the image recognition task, we use the same ResNet-50 but without the temporal dimension for both the input/output data and convolution kernels.

**Training and Testing Settings**   We use MXNet [3] to experiment on the image classification task, and PyTorch [18] on video classification tasks. For image classification, we report standard single model single $224 \times 224$ center crop validation accuracy, following [9, 10]. For experiments on video datasets, we report both single clip accuracy and video accuracy. All experiments are conducted using a distributed K80 GPU cluster and the networks are optimized by synchronized SGD. Code and trained models will be released on GitHub soon.

## 3.2   Ablation Studies

For the ablation studies on Kinetics [1], we use 32 GPUs per experiment with a total batch size of 512 training from scratch. All networks take 16 frames with resolution $112 \times 112$ as input. The base learning rate is set to 0.2 and is reduced with a factor of 0.1 at the 20k-th, 30k-th iterations, and terminated at the 37k-th iteration. We set the number of output channels for three convolution layers $\theta(\cdot)$, $\phi(\cdot)$ and $\rho(\cdot)$ to be $1/4$ of the number of input channels. Note that sub-sampling trick is not adopted for all methods for fair comparison.

**Single Block**   Table 2 shows the results when only one extra block is added to the backbone network. The block is placed after the second residual unit of a certain stage. As can be seen from the last three rows, our proposed $A^2$-block constantly improves the performance compared with both the

Table 2: Comparisons between single nonlocal block [25] and single double attention block on the Kinetics dataset. The performance of vanilla residual networks without extra block is shown in the top row.

| Model | + 1 Block | #Params | FLOPs | $\Delta$ FLOPs | Clip @1 | $\Delta$ Clip@1 | Video@1 |
|---|---|---|---|---|---|---|---|
| ResNet-26 | None | 7.043 M | 8.3 G | – | 50.4 % | – | 60.7 % |
| ResNet-29 | None | 7.620 M | 9.2 G | 900 M | 50.8 % | +0.5 % | 61.6 % |
| ResNet-26 + NL [25] | @ Conv2 | 7.061 M | 49.0 G | 40.69 G | – | – | – |
| | @ Conv3 | 7.112 M | 13.7 G | 5.45 G | 51.5 % | +1.1 % | 62.0 % |
| | @ Conv4 | 7.312 M | 9.3 G | 1.04 G | 51.7 % | +1.3 % | 62.3 % |
| ResNet-26 + $A^2$ | @ Conv2 | 7.061 M | 8.7 G | 463 M | 51.2 % | +0.8 % | 61.8 % |
| | @ Conv3 | 7.112 M | 8.7 G | 463 M | 51.9 % | +1.5 % | 62.0 % |
| | @ Conv4 | 7.312 M | 8.7 G | 463 M | **52.3** % | **+1.9** % | **62.6** % |

Table 3: Comparisons between performance from multiple nonlocal blocks [25] and multiple double attention blocks on Kinetics dataset. We report both top-1 clips accuracy and top-1 video accuracy for all the methods. The vanilla residual networks without extra blocks are shown in the top row.

| Model | +N Blocks | #Params | FLOPs | $\Delta$ FLOPs | Clip @1 | $\Delta$ Clip@1 | Video @1 |
|---|---|---|---|---|---|---|---|
| ResNet-26 | None | 7.043 M | 8.3 G | – | 50.4 % | – | 60.7 % |
| ResNet-29 | None | 7.620 M | 9.2 G | 900 M | 50.8 % | +0.5 % | 61.6 % |
| ResNet-26 + NL [25] | 1 @ Conv4 | 7.312 M | 9.3 G | 1.04 G | 51.7 % | +1.3 % | 62.3 % |
| | 2 @ Conv4 | 7.581 M | 10.4 G | 2.08 G | 52.0 % | +1.6 % | 62.9 % |
| | 4 @ Conv3&4 | 7.719 M | 21.3 G | 12.97 G | 52.4 % | +2.0 % | 62.8 % |
| ResNet-26 + $A^2$ | 1 @ Conv4 | 7.312 M | 8.7 G | 463 M | 52.3 % | +1.9 % | 62.6 % |
| | 2 @ Conv4 | 7.581 M | 9.2 G | 925 M | 52.5 % | +2.1 % | 63.1 % |
| | 4 @ Conv3&4 | 7.719 M | 10.1 G | 1.85 G | **53.0** % | **+2.6** % | **63.5** % |

baseline ResNet-26 and the deeper ResNet-29. Notably the extra cost is very little. We also find that the performance gain from placing $A^2$-block on top layers is more significant than placing it at lower layers. This may be because the top layers give more semantically abstract representations that are suitable for extracting global visual primitives. Comparatively, the Nonlocal Network [25] shows less accuracy gain and more computational cost than ours. Since the computational cost for Nonlocal Network is increased quadratically on bottom stage, we are even unable to finish the training when the block is placed at Conv2.

**Multiple Blocks**   Table 3 shows the performance gain when multiple blocks are added to the backbone networks. As can be seen from the results, our proposed $A^2$-Net monotonically improves the accuracy when more blocks are added and costs less #FLOPs compared with its competitor. We also find that adding blocks to different stages can lead to more significant accuracy gain than adding all blocks to the same stage.

### 3.3   Experiments on Image Recognition

We evaluate the proposed $A^2$-Net on ImageNet-1k [13] image classification dataset, which contains more than 1.2 million high resolution images in $1,000$ categories. Our implementation is based on the code released by [5] using $64$ GPUs with a batch size of $2,048$. The base learning rate is set to $\sqrt{0.1}$ and decreases with a factor of $0.1$ when training accuracy is saturated.

Table 4: Comparison with state-of-the-arts on ImageNet-1k.

| Model | Backbone | Top-1 | Top-5 |
|---|---|---|---|
| ResNet [9] | ResNet-50 | 75.3 % | 92.2 % |
| | ResNet-152 | 77.0 % | 93.3 % |
| SENet [11] | ResNet-50 | 76.7 % | 93.4 % |
| $A^2$-Net | ResNet-50 | **77.0** % | **93.5** % |

Table 5: Comparisons with state-of-the-arts results on Kinetics. Only RGB information is used for input.

| Model | #Frames | FLOPs | Video @1 | Video @5 |
|---|---|---|---|---|
| ConvNet+LSTM [1] | – | – | 63.3 % | – |
| I3D [1] | 64 | 107.9 G | 71.1 % | 89.3 % |
| R(2+1)D [23] | 32 | 152.4 G | 72.0 % | 90.0 % |
| $A^2$-Net | 8 | 40.8 G | **74.6** % | **91.5** % |

Table 6: Comparisons with state-of-the-arts results on UCF-101. The averaged Top-1 video accuracy on three train/test splits is reported.

| Method | Backbone | FLOPs | Video @1 |
|---|---|---|---|
| C3D [21] | VGG | 38.5 G | 82.3 % |
| Res3D [22] | ResNet-18 | 19.3 G | 85.8 % |
| I3D-RGB [1] | Inception | 107.9 G | 95.6 % |
| R(2+1)D-RGB [23] | ResNet-34 | 152.4 G | 96.8 % |
| $A^2$-Net | ResNet-50 | 41.6 G | 96.4 % |

As can be seen from Table 4, a ResNet-50 equipped with 5 extra $A^2$-blocks at Conv3 and Conv4 outperforms a much larger ResNet-152 architecture. We note that the $A^2$-blocks embedded ResNet-50 is also over 40% more efficient than ResNet-152 and only costs 6.5 GFLOPs and 33.0 M parameters. Compared with the SENet [11], the $A^2$-Net also achieves better accuracy which proves the effectiveness of the proposed double attention mechanism.

### 3.4 Experiment Results on Video Recognition

In this subsection, we evaluate the proposed method on learning video representations. We consider the scenario where static image features are pretrained but motion features are learned from scratch by training a model on the large-scale Kinetics [1] dataset, and the scenario where well-trained motion features are transferred to small-scale UCF-101 [20] dataset.

**Learning Motion from Scratch on Kinetics**    We use ResNet-50 pretrained on ImageNet and add 5 randomly initialized $A^2$-blocks to build the 3D convolutional network. The corresponding backbone is shown in Table 1. The network takes 8 frames (sampling stride: 8) as input and is trained for $32k$ iterations with a total batch size of $512$ using $64$ GPUs. The initial learning rate is set to $0.04$ and decreased in a stepwise manner when training accuracy is saturated. The final result is shown in Table 5. Compared with the state-of-the-art I3D [1] and R(2+1)D [23], our proposed model shows higher accuracy even with a less number of sampled frames, which once again confirms the superiority of the proposed double-attention mechanism.

**Transfer the Learned Feature to UCF-101**    The UCF-101 contains about $13,320$ videos from 101 action categories and has three train/test splits. The training set of UCF-101 is several times smaller than the Kinetics dataset and we use it to evaluate the generality and robustness of the features learned by our model pre-trained on Kinetics. The network is trained with a base learning rate of $0.01$ which is decreased for three times with a factor $0.1$, using 8 GPUs with a batch size of 104 clips and tested with $224 \times 224$ input resolution on single scale. Table 6 shows results of our proposed model and comparison with state-of-the-arts. Consistent with above results, the $A^2$-Net achieves leading performance with significantly lower computational cost. This shows that the features learned by $A^2$-Net are robust and can be effectively transferred to new dataset in very low cost compared with existing methods.

## 4   Conclusions

In this work, we proposed a double attention mechanism for deep CNNs to overcome the limitation of local convolution operations. The proposed double attention method effectively captures the global information and distributes it to every location in a two-step attention manner. We well formulated the proposed method and instantiated it as an light-weight block that can be easily inserted into to existing CNNs with little computational overhead. Extensive ablation studies and experiments on a number of benchmark datasets, including ImageNet-1k, Kinetics and UCF-101, confirmed the effectiveness of the proposed $A^2$-Net on both 2D image recognition tasks and 3D video recognition tasks. In the future, we want to explore integrating the double attention in recent compact network architectures [19, 16, 4], to leverage the expressiveness of the proposed method for smaller, mobile-friendly models.

## Footnotes

[2]Here by "space" we mean the entire feature maps of an input frame and the complete spatio-temporal features from a video sequence.

[3]For a spatial (2D) convolution, *i.e.* when the input is an image, $d = 1$.

[4]All values are stored in 32-bit float.

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
