[Reviews · NeurIPS 2018]

Reviewer 1



In this paper, the authors propose a new network structure with 2 step attentions. The structure is inspired by SENet. The first step attention is able to capture second-order statistics, and the second attention adaptively allocates features from a compact bag rather than exhaustively correlating all features. The authors report state-of-art result in image classification and video action understanding. It is an incremental work to SENet. I think the major contribution is on video classification task. Compare to Non-local neural networks, A2 net is able to achieve better result with much less computation. However, my concern is, for video classification task, there is no comparison to ResNet-26 + SENet. There is vanilla residual networks result, but if there is ResNet-26 + SENet result, we will have better understanding on the contribution of first step attention. Then we will have better measurement of the overall contribution. -- Update: I have read the authors' feedback. According to new experiments, I adjusted the score.

Reviewer 2



This paper proposes the “double attention block” to aggregate and propagate informative global features from the entire spatio/spatio-temporal space of input. Specifically, the model first generates a set of attention distributions over the input and obtains a set of global feature vectors based on the attention. Then, for each input position, it generates another attention distribution over the set of global feature vectors and uses this to aggregate those global feature vectors into a position-specific feature vector. The proposed component can be easily plugged into existing architectures. Experiments on image recognition (ImageNet-1k) and video classification (Kinetics, UCF-101) show that the proposed model outperforms the baselines and is more efficient. Strength 1. The proposed module is efficient and easy to adapt to modern frameworks. 2. Extensive empirical experiments show that the proposed method outperforms consistently outperform other baselines. Weakness 1. I am concerned about the originality of the proposed model. The idea of applying an attention over the global feature vectors is conceptually similar to the Transformer architecture (Vaswani et al.). 2. Lack of theoretical analysis of the proposed model. 3. The improvement over the baselines is not significant. For example, in the ablation studies (Table 2 and 3), the relative error reduction over “Non-local neural networks” is less than 2%.

Reviewer 3



This paper proposes a simple attention-based method to capture global features with CNN-based models. The idea is to use attention mechanisms twice where the first one is to generate a set of global features and the second one is to attend the global features at each position. The experimental results are positive as expected. The strength of the paper is the simple and clean formulation. I imagine that this would become a building block of many CNN-based models. The proposed method is well contrasted against other methods. A possible weakness is that it is not extensively compared against SENet [10]. If SENet is good enough for most cases, the significance of the paper decreases since it means that the generalization is not such important in practice. A minor comment: Figure 1 could be improved a bit. There are n and #n which seem to be identical. Also, it looks the depth of Z should be m.